# The Relationship between Macronutrient Distribution and Type 2 Diabetes in Asian Indians

**DOI:** 10.3390/nu13124406

**Published:** 2021-12-09

**Authors:** Amisha Pandya, Mira Mehta, Kavitha Sankavaram

**Affiliations:** Department of Nutrition and Food Science, University of Maryland, College Park, MD 20742, USA; mmehta@umd.edu (M.M.); kavitha@umd.edu (K.S.)

**Keywords:** macronutrient distribution, type 2 diabetes mellitus management, type 2 diabetes in Asian Indians immigrants to the US

## Abstract

Asian Indians (AIs) are at increased risk for type 2 diabetes mellitus than other ethnic groups. AIs also have lower body mass index (BMI) values than other populations, so can benefit from strategies other than weight reduction. Macronutrient distributions are associated with improved glycemic control; however, no specific distribution is generally recommended. This study looks at whether a macronutrient distribution of 50:30:20 (percent of total calories from carbohydrates, fats, and protein) is related to diabetes status in AIs. Diet and Hemoglobin A1c (HbA1c) were assessed from convenience sample of AI adults in Maryland. A ratio of actual to needed calories using the 50:30:20 macronutrient distribution was then tested against diabetes status to identify associations. All groups except non-diabetic females, were in negative energy balance. The non-diabetic group consumed larger actual to needed ratios of protein than pre-diabetics and diabetics. However, all groups consumed protein at the lower end of the Acceptable Macronutrient Distribution Range (AMDR), and the quality of all macronutrients consumed was low. Therefore, weight loss may not be the recommendation for diabetes management for AIs. Increasing protein and insoluble fiber consumption, could play a critical role.

## 1. Introduction

Asian Indians (AIs) in India as well as those who have emigrated to the United States and other western nations are seeing an increase in incidence and prevalence of type 2 diabetes mellitus (T2DM) [1]. Prevalence of T2DM among AIs is estimated at 9.3% across India [2], and 18% in the US [3]. In one Indian State, Gujarat, estimates are 7–14% [4]. Rates of undiagnosed T2DM in AIs in India as well as in other countries is estimated to be approximately 50% [5].

There are complex and poorly understood reasons for the increasing incidence and prevalence of T2DM in AIs, including pathophysiological and sociocultural characteristics specific to this population. AIs have unique physical attributes and cultural attitudes that increase their risk for T2DM compared to other ethnic groups [6]. For example, two commonly cited factors for insulin resistance (IR), a precursor to the development of T2DM, are obesity, as measured by body mass index (BMI), and adverse fat distribution, neither of which seem to be associated with high basal insulin levels in this population [7]. AIs have younger onset for T2DM, and lower BMI values as compared with other populations [1,8,9,10,11]. The age of onset for T2DM in AIs is estimated to occur 10 years earlier than in Europeans, and AIs require lower BMI cut-offs for effective identification of T2DM risk [5]. Additionally, AIs may be predisposed to IR and T2DM because AI children are born smaller, have more fat, and less lean muscle [6]. Reduced lean muscle mass at birth is significant because lean muscle mass contains more mitochondria than fat tissue and so is more metabolically efficient. However, because the secretion of insulin is triggered by adipose tissue upon consumption of food, individuals with higher adipose to lean muscle mass ratios may have increased blood insulin levels, which can trigger a negative feedback response resulting in insulin receptor dysfunction leading to IR and T2DM.

The Asian Indian diet is also high in carbohydrates, and with urbanization and migration, there has been a growing tendency towards processed, refined and higher fat convenience foods, coupled with decreases in physical activity seen both in AIs living in India and abroad [12,13].

The purpose of this study is two-fold: (1) to describe the dietary intake of Asian Indian adults with and without T2DM, and (2) to determine whether there is an association between diabetes status and diet indicative of T2DM. Dietary intake was examined as both dietary quantity as well as dietary quality. Dietary quantity was measured by macronutrients (carbohydrates, fat, and protein) as a proportion of total kilocalories. Dietary quality was indicated by consumption of soluble and insoluble fiber, cholesterol, saturated fat, trans fat, and sugar; both were assessed relative to diabetes status.

## 2. Materials and Methods

### 2.1. Study Population

A convenience sample of 59 AI adults from Mangal Mandir, a Hindu temple in the Baltimore/Washington Metropolitan Area, was taken over a period of three months. Subjects included AI adults ≥ 18 years of age, literate in English, and residents of the US for >5 years. Participants’ demographic information such as age, gender, education, income and number of years in US were collected.

### 2.2. Data Collection

Diabetes status was assessed by hemoglobin A1c readings obtained via physician ordered lab results during a health fair, run by community physicians, and held at the Mangal Mandir temple; an event used to initiate the study. For study participants with no physician’s diagnosis of diabetes or with pre-diabetes, physician ordered labs for fasting blood glucose and hemoglobin A1c were not warranted and therefore were not available. Glucose and HbA1c point of care (POC) monitors were used for those that did not have fasting blood glucose and HbA1c readings from their physicians to confirm diabetes status. Cut-offs used for no-diabetes, pre-diabetes and diabetes are those widely accepted by the American Diabetes Association (ADA), the World Health Organization (WHO), and the International Diabetes Federation (IDF); >5.7% = no diabetes, ≥5.7% but <6.5% = pre-diabetes, and ≥6.5% = diabetes. Hemoglobin A1c level was chosen as the exclusive diagnostic variable because it was the most reliable measure of diabetes status. The collection of fasting blood glucose after 8 h of fasting was found not to be feasible and could not always be obtained from participant physician ordered lab results (about 13% of participants were missing this data point). Therefore, fasting blood glucose was dropped as a diagnostic variable. The accuracy of Bayer’s A1cNow+ monitor was confirmed for non-clinical diagnoses of diabetes with up to 98% agreement, or non-significant difference, between the device and laboratory results [14,15,16,17,18]. Participants also provided self-report of their diabetes status which was compared to their HbA1c to determine the rate of undiagnosed diabetes in this population.

All participants completed a 163-item food frequency questionnaire (FFQ) validated with AIs that gave a one-year retrospective to their dietary intake [19,20]. Diet quantity was measured by the proportion of total kilocalories consumed daily in the form of the three energy producing macronutrients, carbohydrates, fats, and proteins contributing to metabolic efficiency and examined by diabetes status. Diet quality was assessed by the consumption of harmful fats such as saturated and trans-fat, dietary cholesterol, and fiber content of carbohydrates as seen in consumption of insoluble and soluble fiber. Basal metabolic rate (BMR) calculations were performed using five different methods, all varied slightly but they each generally followed the same trend by diabetes status.

The initial selection of BMR equations was based on methods used in various studies. There was consideration given to equations applied to Asian Indians in India, however, their use over other methods was not validated in the literature [21,22,23]. There are four equations commonly cited in the literature. They are the Owen, Mifflin-St. Jeor, Harris Benedict, and the WHO/FAO/UNU; there is varying agreement as to which method is the most reliable [24,25,26,27]. The BMR provided by the Tanita Scale (BC-558 Ironman Segmental Body Composition Monitor) used to capture bioelectrical impedance measures was added to previously mentioned four methods. Each method has its predictive variability, and none can be validated as the most accurate predictor in Asian Indian populations. Thus, all were considered in the analysis and correlational analysis confirmed that all five methods were significantly correlated with one another (*p* < 0.0001). For simplicity, the Owen method was selected for the remainder of the analyses. BMR establishes the minimum caloric intake needed to meet energy requirement assuming no physical activity. Energy requirements for each participant were calculated (BMR + kilocalories burned through physical activity).

### 2.3. Macronutrient Distribution

Although the United States Institute of Medicine’s (IOM’s) Acceptable Macronutrient Distribution Ranges (AMDR) offers recommended ranges of macronutrient intake, no specific recommendations exist to achieve optimal metabolic efficiency, however, there is some evidence that suggests that a macronutrient distribution of 50:30:20 (percent of total calories from carbohydrates, fats, and protein) can be metabolically favorable [28,29,30,31]. To establish a reference value for macronutrient distribution, total caloric energy requirement as determined from BMR + kilocalories burned through physical activity, was multiplied by a standard recommended proportion of total kilocalories for each macronutrient (50 percent of total kilocalories from carbohydrates, 30 percent of total kilocalories from fat, and 20 percent of total kilocalories from protein) to obtain the daily required grams of each macronutrient. A ratio of actual to needed kilocalories from each macronutrient was then calculated by dividing daily intake of macronutrients (carbohydrate, fat, and protein) by the daily required grams of each macronutrient. These ratios of actual to needed carbohydrates, fats and proteins were then tested against diabetes status to identify associations.

### 2.4. Statistical Analysis

Descriptive statistics such as mean, standard deviations and correlational analyses were used to describe the data and establish covariance for any variables. Univariate analyses of variance, including *t*-tests, ANOVAs, and linear and logistic regression models were used to determine the association of diet across diabetes status groups. Multiple linear logistic regression was used to determine the relationship between diet and diabetes status. Those participating as married couples, warranted the data being examined to determine any effect this may have had on variable confounding. A correlational analysis was performed between males and females of participant couples.

## 3. Results

### 3.1. Participant Demographics

Fifty-nine individuals initially expressed interest in participating in the study, and 39 participants completed the study (power = 0.76). Study participants were about equally divided by gender (49% male and 51% female) with an average age of 65.2 years (67.4 years for males and 63.0 years for females). About 72% of participants were married. Participants were predominantly Gujarati immigrants (95%) who had lived an average of 37 years in the United States (39 years for males and 35 years for females). Ninety-seven percent of all participants were born in India or Africa, and 3% were born in the United States. Almost two thirds (61.5%) of participants had earned a bachelor’s degree or less and 38.5% had earned post graduate or professional degrees. Participants were almost equally split in terms of household income, with 56.4% earning less than or equal to USD 100,000 annually and 43.6% earning more than USD 100,000 annually. Almost three quarters of participants were vegetarian (58% of males and 90% of females, *p* < 0.0310). About 90% of participants, both male and female, consumed alcohol occasionally (1–2 times/month) or never, and most participants did not have a history of smoking (92% overall, 100% for females, and 84% for males). About 53% of males and 70% of females reported primarily being responsible for grocery shopping in their households, whereas most females were predominantly responsible for cooking (0% of males, and 90% of females, *p* < 0.0001).

### 3.2. Diabetes Status

Participant self-report was compared to HbA1c groupings for both diabetes and pre-diabetes to determine the rate of undiagnosed diabetes in this population. Although, 71% of participants correctly self-reported having diabetes and 88% self-reported having pre-diabetes, there were 12% undiagnosed for diabetes and 39% undiagnosed for pre-diabetes. Overall, there were 39% undiagnosed for diabetes and prediabetes, by gender that broke down to be 26% of males and 50% of females.

Participants were 18% non-diabetic (10% male and 8% female), 49% were pre-diabetic (13% male and 36% females), and 33% were diabetic (26% male and 8% female); the difference by gender across these three diabetes status groups is significant (F = 0.0017, *p* = 0.0197). Figure 1 shows the breakout of diabetes status group by gender.

### 3.3. Participant Diets

Participants consumed an average of 1463.7 kilocalories per day (1600.9 for males and 1333.3 for females, *p* = 0.01). Figure 2 shows the macronutrient intake per day (in grams) for all participants as well as for males and females separately. Daily consumption of protein averaged 51.4 g across all participants (46.8 g for females and 56.4 g for males), carbohydrates averaged 201.3 g across all participants (181.3 g for females and 222.4 g for males), and fats averaged 49.5 g across all participants (46.4 g for females and 52.9 g for males).

Among married participants, couples consumed similar diets, however, participants were not significantly correlated on diabetes status. Therefore, comparisons by diabetes status or gender would still reveal real differences between Asian Indian male and female non-diabetics, pre-diabetics and diabetics.

Although males and females consumed significantly different amounts of total kilocalories per day, they did not differ significantly when each macronutrient was examined as the percent of total kilocalories, or when macronutrient components were examined as proportion of their respective macronutrient. Protein, as a percent of total kilocalories, was about 14% for both males and females, carbohydrates, as a percent of total kilocalories, was 56% for males and 55% for females, and fat, as a percent of total kilocalories, was 29% for males and 31% for females.

Although the percentages of total kilocalories from protein, carbohydrates, and fats for both males and females fell within acceptable macronutrient distribution ranges for daily consumption by adults in the US as recommended by the Food and Nutrition Board, Institute of Medicine, National Academies [32], protein as percent of total kilocalories was at the lower recommended ranges for daily consumption. Table 1 provides a comparison of the IOM’s AMDR with participant daily intakes. Participants had lower than recommended daily ranges for total fiber consumption (16.10 g for females and 19.00 g for males, IOM recommendations ranged 21–25% for females and 30–38% for males) and dramatically less than the maximum allowances for sugar (<1% for both males and females; IOM maximum allowance is 25%). No recommendations were given by the IOM for monounsaturated fatty acids and the nutrient data for this study did not break out polyunsaturated fatty acids into n-6 (linoleic acid) and n-3 (α-linolenic acid), so those comparisons were not possible. Saturated fat and trans fatty acids were consumed in small quantities by participants (<1% and <<1% respectively), and cholesterol was consumed in relatively small quantities as well (5.35% of total kilocalories for females, and 6.63% for males).

Looking at these dietary components by diabetes status reveals that there was a significant difference between the proportion of trans fat consumed of total fat across groups. Non-diabetics consumed the largest proportion (1%), pre-diabetics consumed a smaller proportion (0.2%), and diabetics consumed the smallest proportion of all (0.08%), *p* = 0.036*. There was a difference in the consumption of cholesterol as a proportion of total fat, but it was not significant (*p* = 0.2225). Non-diabetics again consumed the largest proportions (2.67) followed by diabetic (1.85), and then pre-diabetic (1.41), *p* = 0.0636. These differences were not seen when looking at males alone, but were when looking at females alone, albeit not significantly. Non-diabetic females consumed 1% of total fat from trans-fat, whereas pre-diabetic consumed 0.1%, and diabetic consumed 0.03%, *p* = 0.0522. For proportion of cholesterol, non-diabetic females consumed the most (3.03), followed by pre-diabetic females (1.40), and then diabetic females (0.74), *p* = 0.0867.

### 3.4. BMR by Diabetes Status

Table 2 provides the mean BMR by gender and diabetes status. BMR establishes the minimum caloric intake needed to meet energy requirement assuming no physical activity. Energy requirements for each participant were calculated (BMR + kilocalories burned through physical activity).

Figure 3, Figure 4 and Figure 5 show the average BMR, average caloric intake needed to meet energy requirements and average caloric intake by diabetes status and gender. Correlational analysis shows that BMR is significantly correlated with total caloric intake for all participants (0.4251, *p* = 0.0070), but not for males only (0.0334, *p* = 0.8920), or females only (0.1088, *p* = 0.6478). However, BMR was significantly correlated with total caloric intake needed to meet energy requirements for all participants, males only and females only (0.7799, *p* < 0.0001; 0.6738, *p* = 0.0016; 0.6865, *p* = 0.0008, respectively). Total caloric intake to meet energy needs significantly exceeds total kilocalories consumed for all participants as well as for males only and females only (0.3015, *p* = 0.0625; 0.0444, *p* = 0.8569; 0.2550, *p* = 0.2779, respectively), except for female non-diabetics, whose caloric intake exceeds total kilocalories needed for energy needs.

### 3.5. Diet Quantity—Macronutrient Distributions—Actual to Needed Calorie Ratios

Kilocalories required to meet energy needs exceeded caloric intake for all groups, except non-diabetic females; caloric intakes for diabetic females were slightly higher than what is needed to meet energy needs.

The association between macronutrient distribution and diabetes status was examined by looking at the ratio of actual to needed total kilocalories, protein, carbohydrates and fats based on energy needs by gender and diabetes status (Figure 6, Figure 7, Figure 8 and Figure 9).

There were no significant differences noted by diabetes status for total actual to needed kilocalories (Figure 6). However, non-diabetic females did exceed their total needed caloric intake (105%), whereas pre-diabetic and diabetic females consume 81% and 80%, respectively, of their needed kilocalories. Actual to needed ratios of total kilocalories were lower for non-diabetic men (78%) than pre-diabetic (81%) and diabetic men (80%).

Figure 7 shows non-diabetics consuming higher ratios of actual to needed protein than pre-diabetics and diabetics (64%, 59% and 60%, respectively); this difference was not statistically significant (*p* = 0.8450). Non-diabetic females consumed higher ratios of actual to needed protein than pre-diabetic and diabetic females (83%, 61%, and 57%, respectively); this difference was not statistically significant (*p* = 0.0699). Male diabetics consumed the highest ratios of actual to needed protein (61%), followed by pre-diabetics (55%) and then non-diabetics (50%); this difference was not statistically significant (*p* = 0.6786).

Correlational analysis also showed a significant relationship between diabetes status and actual to needed kilocalories from protein for females (−0.45, *p* = 0.0458). This relationship was not seen in males; however, correlational analysis did show a significant relationship for males only between HbA1c level and actual to needed kilocalories from protein (0.51, *p* = 0.0243); this relationship is in the reverse direction than seen in females.

Figure 8 shows a similar pattern for actual to needed kilocalories from carbohydrates as shown for actual to needed consumption of protein. Non-diabetic females were consuming higher ratios of carbohydrates than pre-diabetics (111% and 96%, respectively); diabetic females consumed the lowest ratios of carbohydrates (90%). The pattern for men is slightly different. Pre-diabetic males consumed the highest ratio of actual to needed carbohydrates (94%), followed by non-diabetic (88%) and finally diabetic males (87%).

Figure 9 shows actual to needed consumption of fat was highest for non-diabetic females, exceeding the daily caloric needed for fat (111%). Pre-diabetics consumed the second highest percent (91%), followed by diabetics (86%). For males, however, diabetics consumed the highest percent of actual to needed fat (81%), followed by non-diabetics (80%) and then pre-diabetics (77%).

### 3.6. Diet Quality

Additional findings from the correlational analysis showed a significant relationship between diabetes status and proportion of trans fats (−0.34034, *p* = 0.034) and a weak correlation between diabetes status and proportion of insoluble fiber (0.30208, *p* = 0.0616) for all participants, and a significant correlation between diabetes status and proportion of insoluble fiber (0.45653, *p* = 0.043) for females only. However, when correlations between HbA1c, macronutrients and other dietary components were examined, percent protein (0.33328, *p* = 0.0381), proportion of soluble fiber (0.31638, 0.0497), and total kilocalories consumed were significantly correlated with HbA1c levels for all participants. Males did not show any specific significant correlations between diabetes status and any macronutrient, however, when looking at the correlations between HbA1c, macronutrients and other dietary components, percent protein (0.50846, *p* = 0.0262), and proportion of insoluble fiber (0.50401, *p* = 0.0278), were significantly correlated, while proportion of soluble fiber was weakly correlated with diabetes status for males (0.45357, *p* = 0.0511).

### 3.7. Predicting Diabetes Status

A multiple linear regression model was performed to test how well macronutrient independent variables, that were correlated with diabetes status, could predict diabetes status. With diabetes status as the dependent variable the initial model included actual to needed kilocalories from protein, actual to needed kilocalories from carbohydrates, actual to needed kilocalories from fat, proportion of trans fat, proportion of cholesterol, proportion of soluble and insoluble fiber, percent protein as independent variables. The resulting model was significant (F = 2.58, *p* = 0.0282). All independent variables were significant predictors of diabetes status, except proportion of trans fat and proportion of soluble fiber. However, when these variables were removed from the model, the model was no longer significant, so these variables were retained in the overall model. Table 3 provides the test statics for this multiple regression model. Table 4 shows the results when the same independent variables were tested to predict HbA1c level, the overall model was also significant (F = 4.39, *p* = 0.0013). The model’s performance was improved by removing proportion of cholesterol from the model. The result was a significant regression for the remaining variables (F = 2.66, *p* = 0.0012), with an R2 of 0.4165.

## 4. Discussion

This study established an association between diet as measured by actual to needed macronutrients and diabetes status. Pre-diabetics and diabetics consumed lower ratios of actual to needed protein, carbohydrates and fats relative their energy requirements than non-diabetics. This is consistent with recommendations normally given to patients with IR, pre-diabetes or diabetes; eat less to lose weight [33]. In addition, participants with pre-diabetes and diabetes ate smaller proportions of insoluble and higher proportions of soluble fiber. Consistent with typical AI diets, refined carbohydrates are preferred over whole grains, however, non-diabetics consumption of higher amounts of insoluble fiber may suggest that carbohydrate quality may be beneficial in this population.

This study also shows that Asian Indian pre-diabetics and diabetics recognize the benefit of physical activity and caloric restriction, as they relate to their diabetes status and as recommended for diabetes management by health care providers; and establishes a pattern of decreased overall caloric intake, and decreased intake for each macronutrient (protein, carbohydrate and fat), below required levels based on energy requirements for pre-diabetics and diabetics in the United States. Additionally, it is not known if the combination of increased physical activity and decreased caloric intake discovered in this population can be viewed as contributing to health or negatively impacting diabetes status.

Understanding associations between diabetes status and dietary intake related to diabetes may provide evidenced based strategies to reduce risk of T2DM in this population. As would be the norm in traditional Indian families, females in this population were responsible for cooking and had control over dietary consumption for themselves as well as their families. Additionally, as 72% of participants were married couples, it was noted that of those participants, 70% of the females did not have diabetes or had pre-diabetes while their spouses had diabetes. Along with females being responsible for the household cooking, this suggests a level of control over dietary options that may have an impact on their spouses’ diabetes status.

Contrary to the normal pattern of low physical activity among AIs, levels of physical activity by diabetes status in this study population suggest participants may be recognizing the need for physical activity as a strategy to manage diabetes.

These findings suggest that although AIs may be implementing strategies typically recommended to patients with IR, pre-diabetes and diabetes, those strategies may not be improving disease status in this population. Perhaps a focus on macronutrient balance with sufficient caloric intake to meet energy needs, with increased protein intake, and reduced intake of highly refined carbohydrates in this population would be more effective. A recent comparison of various diets, such as the Mediterranean, Dietary Approaches to Stop Hypertension (DASH), plant-based, low and very low carbohydrate, low-fat, and high protein, aimed at improving metabolic syndrome, T2DM, Cardiovascular Disease (CVD), and hypertension, illustrates that a balance of both macronutrient proportions and quality are needed. In particular, it can be inferred that the macronutrient proportions studied here (50:30:20) provide a starting point from which minor adjustments can be made for individuals’ management of T2DM. In addition, the quality of those macronutrients play an important role (e.g., whole grains as compared to processed carbohydrates, fresh fruits and vegetables, plant-based proteins, and unsaturated fats) [31,34]. Additionally, because carbohydrates and fats are primarily implicated in metabolic disorders, studies vary fat and carbohydrate intake (high fat: low carbohydrate vs. low fat: high carbohydrate) and hold protein intake constant. These studies generally show little difference between these two intake conditions [35]. As suggested in this study, in terms of macronutrient proportions, the key may be in moderating carbohydrates and fats, while adjusting protein intake, as well as ensuring quality of macronutrients consumed. Similarly, one additional study looking at regional differences in dietary patterns in Asian Indians in India, suggests that this strategy may be the most beneficial in the management of T2DM [36].

Notable limitations of this study include that generalization of findings to all AIs, or Gujaratis, was not possible because this study used a convenience sample, and most participants were from a specific cohort of AIs; first generation older Gujarati adults with high levels of education, income and acculturation. Additionally, the cross-sectional design of the current study only allowed for examination of associations between diet and diabetes status, and do not imply causality.

Notable strengths of this study include that this is the first study to examine and identify a relationship between diabetes status and macronutrient composition; suggesting a greater adherence to an optimal macronutrient composition (50:30:20; for percent of total kilocalories from carbohydrates, fat and protein) in non-diabetics than in pre-diabetics or diabetics.

The prevalence of type 2 diabetes and its associated co-morbidities within AI populations, especially Gujaratis, in the US and elsewhere is at extremely high levels and is continuing to rise. As such, further research in this population is needed to more thoroughly study the relationship between diet quality and diabetes status to develop evidence-based strategies helpful in the prevention and management of diabetes in this high-risk population. Specifically, further examination is needed on whether the 50:30:20 macronutrient distribution with caloric intakes consistent with energy needs and improved macronutrient quality has an impact on diabetes status and whether adjustments to diet in these ways could impact diabetes related outcomes.

## 5. Conclusions

AIs are at high risk for insulin resistance (IR) leading to impaired glucose tolerance (IGT) and T2DM and their sequelae. Future research is needed to establish the association between diet quality and diabetes status; however, the findings of this study suggest that those at risk for T2DM may benefit from adhering to a macronutrient distribution of approximately 50:30:20 percent of total kilocalories from carbohydrates, fat and protein, or from a higher intake of dietary protein, while ensuring the quality of those macronutrients. Finally, the implications of near universal female responsibility for cooking, in households where a spouse or other family members are diagnosed with either pre-diabetes or diabetes, are intriguing. Targeted education to AI females about preparing meals that adhere to optimal dietary choices, may have a potential benefit to the entire family including those with diabetes and prediabetes.

The current study is the first to investigate a relationship between the consumption of macronutrients in general and in specific relative proportion (50:30:20) and diabetes status in an AI population. These relationships can be further explored to develop a recommended diet for AIs that is quantitatively and qualitatively supportive metabolically for those with IR, pre-diabetes or diabetes.

## 6. Study Limitations

This study lacked sufficient funding to enable laboratory testing to establish hemoglobin A1c levels for group assignment. Therefore, the study leveraged an annual health fair event at the data collection site which provided for this laboratory testing at no cost to the participants. In cases where study participants did not participate in the health fair or their diabetes status did not indicate laboratory testing by a physician, Hemoglobin A1c levels were supplemented by use of the Bayer A1cNow+ POC monitor for group assignment by diabetes status.

## Figures and Tables

**Figure 1 nutrients-13-04406-f001:**
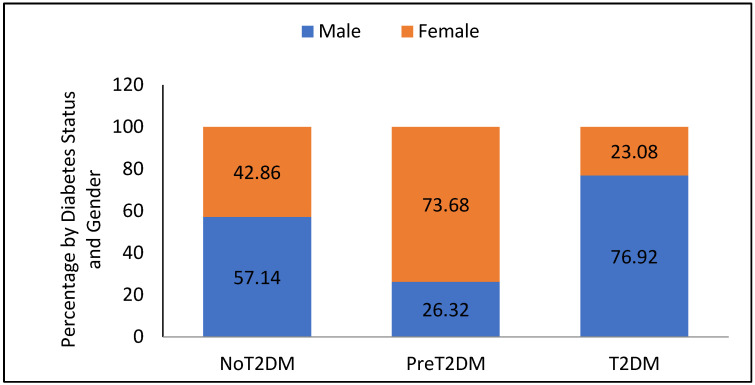
Participants by diabetes status and gender (T2DM = Type 2 Diabetes Mellitus).

**Figure 2 nutrients-13-04406-f002:**
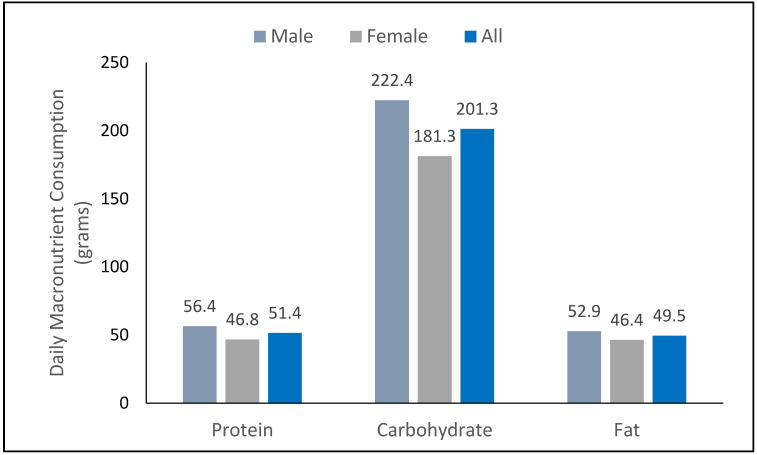
Daily macronutrient consumption.

**Figure 3 nutrients-13-04406-f003:**
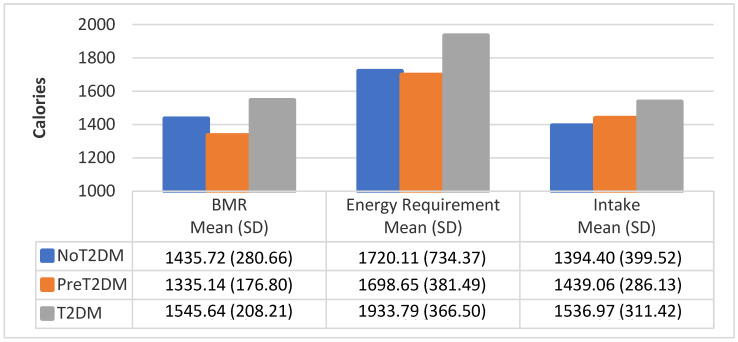
BMR, energy requirement, and caloric intake for all participants by diabetes status.

**Figure 4 nutrients-13-04406-f004:**
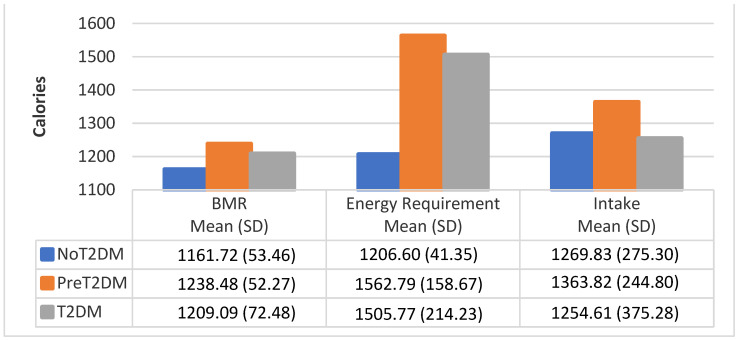
BMR, energy requirement, and caloric intake for female participants by diabetes status.

**Figure 5 nutrients-13-04406-f005:**
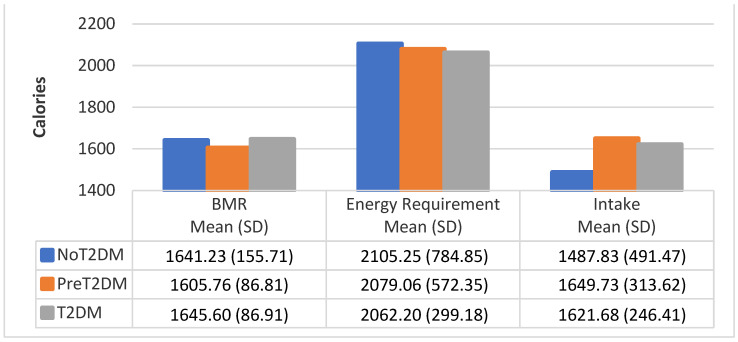
BMR, energy requirement, and caloric intake for male participants by diabetes status.

**Figure 6 nutrients-13-04406-f006:**
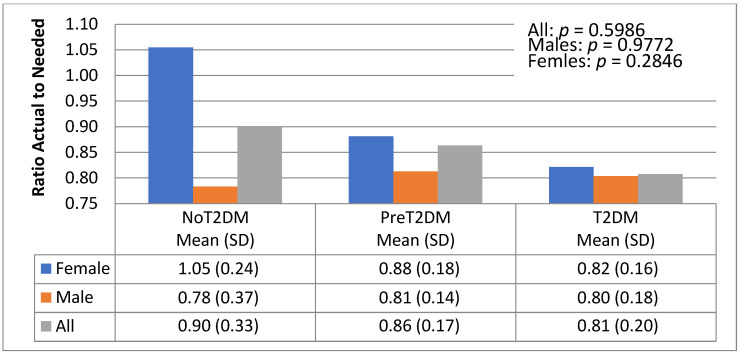
Actual to needed total kilocalories by diabetes status and gender.

**Figure 7 nutrients-13-04406-f007:**
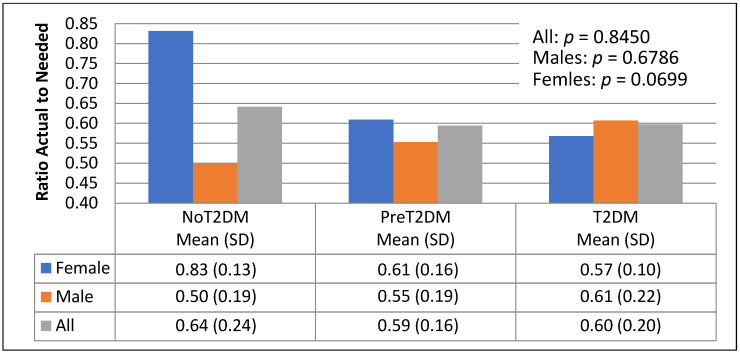
Actual to needed kilocalories from protein by diabetes status and gender.

**Figure 8 nutrients-13-04406-f008:**
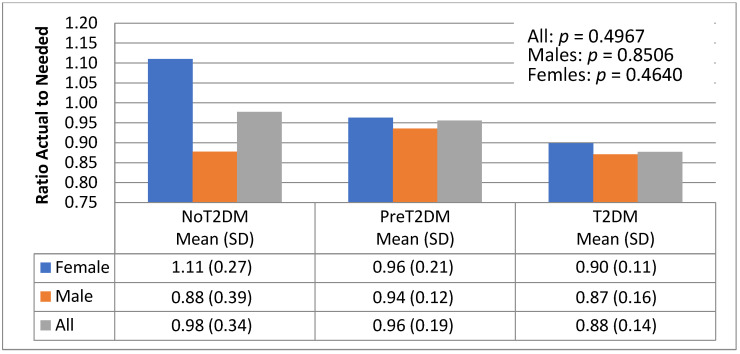
Actual to needed kilocalories from carbohydrates by diabetes status and gender.

**Figure 9 nutrients-13-04406-f009:**
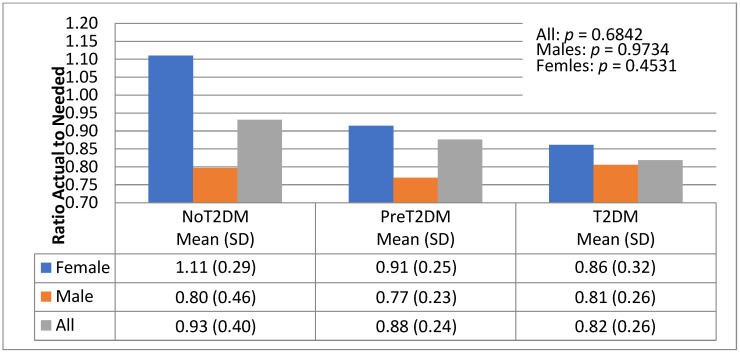
Actual to needed kilocalories from fat by diabetes status and gender.

**Table 1 nutrients-13-04406-t001:** Dietary Components as Compared to IOM’s Acceptable Macronutrient Distribution Ranges (AMDR).

Variable	IOM Males	IOM Females	Female (*n* = 20)	Male (*n* = 19)
Protein as Percent of Total Kilocalories	10–35%	10–35%	14.1%	14.0%
Carbohydrates as Percent of Total Kilocalories	45–65%	45–65%	54.5%	56.0%
Fat as Percent of Total Kilocalories	20–35%	20–35%	31.0%	29.3%
Total Fiber	30–38 g	21–25 g	16.1 g	19.0 g
Soluble Fiber	NA	NA	7.3g	8.6g
Insoluble Fiber	NA	NA	7.4g	9.5g
Sugar	25%	25%	0.76%	0.87%
Monounsaturated Fat	NA	NA	1.4%	1.3%
Polyunsaturated Fat	NA	NA	0.77%	0.69%
n-6 (linoleic acid)	5–10%	5–10%	NA	NA
n-3 (α-linolenic acid)	0.6–1.2%	0.6–1.2%	NA	NA
Saturated Fat	Minimal	Minimal	12.7g (0.94%)	14.8g (0.91%)
Trans Fat	Minimal	Minimal	0.20g (0.01%)	0.19g (0.01%)
Cholesterol	Minimal	Minimal	68.9g (5.35%)	106.4g (6.63%)

IOM: Institute of Medicine.

**Table 2 nutrients-13-04406-t002:** Basal Metabolic Rate (BMR) Calculated Using Five Different Methods by Gender and Diabetes Status.

BMR Method	No-Diabetes	Pre-Diabetes	Diabetes
All	(*n* = 7)	(*n* = 19)	(*n* = 13)
	Mean (SD)	Mean (SD)	Mean (SD)
Owen	1435.73 (280.7)	1335.14 (176.8)	1545.64 (208.2)
Harris-Benedict	1359.18 (300.2)	1291.30 (140.0)	1411.39 (186.9)
Mifflin St. Jeor	1286.67 (307.4)	1192.23 (193.9)	1381.08 (203.2)
WHO	1253.27 (177.0)	1276.27 (106.9)	1362.19 (116.2)
Tanita Scale	1347.50 (303.0)	1252.87 (183.2)	1455.81 (234.7)
Females	(*n* = 3)	(*n* = 14)	(*n* = 3)
Owen	1161.72 (53.5)	1238.49 (52.3)	1209.09 (72.5)
Harris-Benedict	1188.51 (157.0)	1238.88 (96.7)	1198.90 (107.4)
Mifflin St. Jeor	1049.64 (193.2)	1103.50 (126.8)	1086.58 (129.0)
WHO	1147.99 (110.2)	1248.83 (99.2)	1247.90 (136.7)
Tanita Scale	1101.83 (111.8)	1164.61 (94.3)	1094.50 (119.0)
Males	(*n* = 4)	(*n* = 5)	(*n* = 10)
Owen	1641.23 (155.7)	1605.76 (86.8)	1646.60 (86.9)
Harris-Benedict	1487.18 (.335.9)	1438.10 (145.7)	1475.14 (156.3)
Mifflin St. Jeor	1464.45 (256.6)	1440.68 (110.2)	1469.43 (117.5)
WHO	1332.23 (187.5)	1353.08 (96.8)	1396.47 (90.4)
Tanita Scale	1531.75 (263.9)	1500.00 (136.1)	1564.20 (117.1)

**Table 3 nutrients-13-04406-t003:** Parameter Estimates for Predicting Diabetes Status.

Variable	DF	Parameter Estimate	Standard Error	t Value	Pr > |t|	Standardized Estimate
Intercept	1	8.24	2.74	3.01	0.01 *	0.00
Ratio of Actual to Needed Kilocalories from Protein	1	9.68	3.75	2.58	0.02 *	2.51
Ratio of Actual to Needed Kilocalories from Carbohydrates	1	−4.60	1.68	−2.73	0.01 *	−1.35
Ratio of Actual to Needed Kilocalories from Fat	1	−2.18	1.02	−2.14	0.04 *	−0.86
Proportion of Trans Fat (of Total Fat)	1	−39.61	19.41	−2.04	0.05	−0.32
Proportion of Cholesterol (of Total Fat)	1	−0.19	0.10	−1.91	0.07	−0.33
Proportion of Soluble Fiber (of Total Carbohydrates)	1	−61.04	29.21	−2.09	0.04 *	−0.75
Proportion of Insoluble Fiber (of Total Carbohydrates)	1	60.82	22.42	2.71	0.01 *	0.91
Percent Protein (of Total Kilocalories)	1	−0.45	0.18	−2.5	0.02 *	−1.43

* Indicates a significant *p*-value (*p* < 0.05).

**Table 4 nutrients-13-04406-t004:** Parameter Estimates for Predicting A1c Level.

Variable	DF	Parameter Estimate	Standard Error	t Value	Pr > |t|	Standardized Estimate
Intercept	1	14.41	3.39	4.25	<<0.01 *	0
Ratio of Actual to Needed Kilocalories from Protein	1	15.50	4.73	3.28	<0.01 *	2.74
Ratio of Actual to Needed Kilocalories from Carbohydrates	1	−6.54	2.10	−3.12	<0.01 *	−1.31
Ratio of Actual to Needed Kilocalories from Fat	1	−3.37	1.31	−2.57	0.01 *	−0.90
Proportion of Trans Fat (of Total Fat)	1	−61.91	25.17	−2.46	0.02 *	−0.34
Proportion of Soluble Fiber (of Total Carbohydrates)	1	−57.59	34.42	−1.67	0.10	−0.49
Proportion of Insoluble Fiber (of Total Carbohydrates)	1	75.27	27.56	2.73	0.01 *	0.77
Percent Protein (of Total Kilocalories)	1	−0.64	0.23	−2.75	0.01 *	−1.37

* Indicates a significant *p*-value (*p* < 0.05); < indicate *p* < 0.005; << indicates *p* < 0.0005.

## Data Availability

The data presented in this study are available upon request from the corresponding author.

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
