# Peer review of "The Relationship between Macronutrient Distribution and Type 2 Diabetes in Asian Indians"

_nutrients, 2021, doi:10.3390/nu13124406_

Round 1

Reviewer 1 Report

The article raises the interesting issue of the relationship between diabetes status and diet indicative of T2DM. The results obtained in the study can be significant from the point of view of further development of a recommended diet for Asian Indians that may  metabolically support patients with insulin resistance, pre-diabetes or diabetes.

However, I have some concerns which are as follows:

Abstract, lines 16 - 18: the authors state that "Non-diabetics consumed larger actual to needed ratios of protein ...." and then "Protein was consistently consumed at the lower end of the recommended range...." - who consumed protein at the lower end of the recommended range?

Material and Methods

Study population: This is not clear for me, it seems like there were two locations of samples collection.

Data collection: In my opinion it is difficult to combine results obtained using two different methods, i.e., Hemoglobin A1c readings from physicians and HbA1c point of care. There is also information missing when did the patients have HbA1c tested? This also may change with time, and the actual health status can thus be different.

Results: line 132 - fifty-nine patients - the Methods part indicated 60

Section 3.5: results concerning insoluble and solube fiber are not presented, only total fiber is included in Table 1

Section 3.6: There are no results of proportion of vegetable protein and proportion of carbohydrates absent any fiber

Tables 3 and 4 are missing.

Discussion, line 379 - I can't find any information about antrophometric measurements in the Methods part, nor results of such measurements.

Other comments:

When citing literature, dots should be after parentheses. 

line 88: BMR - should be explained when used for the first time

line 107: IOM - should be explained

e.g. line 143,a dn in the whole text: p<0.0310 instead of p<.0310

line 220: "(note lines are used for visual comparison of the points)" - I think this information is redundant

line 242: comment like above

Figures 7 - 9: "mean (SD)" shoule be added like in other figures

I think the footnotes are redundant (page 1, 2 and 3)

Author Response

Abstract, lines 16 - 18: the authors state that "Non-diabetics consumed larger actual to needed ratios of protein ...." and then "Protein was consistently consumed at the lower end of the recommended range...." - who consumed protein at the lower end of the recommended range?

Response: Non-diabetics in this study consumed larger actual to needed ratios of protein, however, the entire study population consumed protein at the lower range of the AMDR as published by the IOM and provided in Table 1. Reworded to read, "The non-diabetic group consumed larger actual to needed ratios of protein than pre-diabetics and diabetics. However, all groups consumed protein at the lower end of the AMDR range, and the quality of all macronutrients consumed was low."

Material and Methods

Study population: This is not clear for me, it seems like there were two locations of samples collection.

Response: the study population section was reworded to reflect that there was only one site for data collection, the Mangal Mandir, a Hindu temple in the Baltimore Washington Metropolitan Area; Maryland is a state within that area.  It now reads, "A convenience sample of 59 AI adults from Mangal Mandir, a Hindu temple in the Baltimore/Washington Metropolitan Area, was taken over a period of three months." 

Data collection: In my opinion it is difficult to combine results obtained using two different methods, i.e., Hemoglobin A1c readings from physicians and HbA1c point of care. There is also information missing when did the patients have HbA1c tested? This also may change with time, and the actual health status can thus be different.

Response: the first 3 sentences of the data collection section have been reworded.  Physician order lab test were ordered and obtained in the same 3 month data collection period as POC tests were completed.  POC tests were primarily used for participants that did not have physician ordered fasting glucose or HbA1c because those tests were not indicated for the patient by the physician.  It now reads, "Diabetes status was assessed by Hemoglobin A1c readings obtained via physician ordered lab results during a health fair, run by community physicians, and held at the Mangal Mandir temple; an event used to initiate the study.  For study participants with no physician’s diagnosis of diabetes or with pre-diabetes, physician ordered labs for fasting blood glucose and hemoglobin A1c were not warranted and therefore were not available.  Glucose and HbA1c point of care (POC) monitors were used for those that did not have fasting blood glucose and HbA1c readings from their physicians to confirm diabetes status."

Also added are literature references for the comparability of capillary HbA1c POC monitors with venous blood draws analyzed within a lab. And added the following text, "The accuracy of Bayer’s A1cNow+ monitor was confirmed for non-clinical diagnoses of diabetes with up to 98% agreement, or non-significant difference, between the device and laboratory results."

Additionally, a limitations section has been added to further explain why lab tests were not ordered for those without physician ordered lab tests from the health fair.

Results: line 132 - fifty-nine patients - the Methods part indicated 60

Response: The target convenience sample was 60.  59 expressed interest.  I have changed the study participant section above to reflect the convenience sample was 59 and not 60.

Section 3.5: results concerning insoluble and soluble fiber are not presented, only total fiber is included in Table 1

Response: To my knowledge the IOM does not publish soluble and insoluble fiber AMDR values nor are there RDA values for either, only Total Fiber.  I have included both in Table 1 indicating that AMDR values are not applicable (NA) from the IOM.

Section 3.6: There are no results of proportion of vegetable protein and proportion of carbohydrates absent any fiber

Response: the two variables you reference were not presented in the paper because they were 2 of many variables considered but not significantly meaningful.  I have removed the reference to those two variables.  The section (lines 305-309) now reads, "With diabetes status as the dependent variable the initial model included actual to needed kilocalories from protein, actual to needed kilocalories from carbohydrates, actual to needed kilocalories from fat, proportion of trans fat, proportion of cholesterol, proportion of soluble and insoluble fiber, percent protein as independent variables. The resulting model was significant (F=2.58, p=0.0282)."

Tables 3 and 4 are missing.

Response: My apologies.  They have been re-inserted.

Discussion, line 379 - I can't find any information about anthropometric measurements in the Methods part, nor results of such measurements.

Response: Apologies, that is the subject of another paper.  My study looked at a few different factors associated with diabetes, including anthropometric measurements.  However, this paper focuses on macronutrient distribution.

Other comments: When citing literature, dots should be after parentheses.

Response: corrected to have periods after citation brackets ([  ].)

line 88: BMR - should be explained when used for the first time

Response: wrote out basal metabolic rate when first used on line 95.

line 107: IOM - should be explained e.g. line 143,a dn in the whole text: p<0.0310 instead of p<.0310

Response: spelled out United States Institute of Medicine’s (IOM’s) on line 114.  Also corrected all instances with p<. to p<0.

line 220: "(note – lines are used for visual comparison of the points)" - I think this information is redundant

Response: deleted note - graphs were changed to histograms from lines.

line 242: comment like above

Response: removed the text "lines are used for visual comparison of the points" as line graphs were replaced by histograms so no longer needed/relevant.

Figures 7 - 9: "mean (SD)" should be added like in other figures

Response: Added Mean (SD) labels on data tables for figures 7-9.

I think the footnotes are redundant (page 1, 2 and 3)

Response: removed footnotes from pages 1, 2 and 3.

Reviewer 2 Report

The topic of this work is relevant, since diabetes mellitus is a major social problem. The study is valuable as it is also highlighting the data collection of the people literate in English and more than 18 years old. Also, this article highlights that weight loss may not be the recommendation for diabetes management. But diet rich with protein and fiber play an important role. 

Author Response

I have addressed all comments from the reviewer.

Round 2

Reviewer 1 Report

Thank you for the revised manuscript. I am satisfied with the authors' corrections and clarifications.